# Production, Composition and Nutritional Properties of Organic Milk: A Critical Review

**DOI:** 10.3390/foods13040550

**Published:** 2024-02-11

**Authors:** Kevin Linehan, Dhrati V. Patangia, Reynolds Paul Ross, Catherine Stanton

**Affiliations:** 1Teagasc Food Research Centre, Moorepark, Fermoy, P61 C996 Cork, Ireland; 2APC Microbiome Ireland, University College Cork, T12 Y120 Cork, Ireland; 3School of Microbiology, University College Cork, T12 XF62 Cork, Ireland; 4VistaMilk Research Centre, Teagasc Moorepark, Fermoy, P61 C996 Cork, Ireland

**Keywords:** organic, milk, dairy, composition, milk production systems

## Abstract

Milk is one of the most valuable products in the food industry with most milk production throughout the world being carried out using conventional management, which includes intensive and traditional systems. The intensive use of fertilizers, antibiotics, pesticides and concerns regarding animal health and the environment have given increasing importance to organic dairy and dairy products in the last two decades. This review aims to compare the production, nutritional, and compositional properties of milk produced by conventional and organic dairy management systems. We also shed light on the health benefits of milk and the worldwide scenario of the organic dairy production system. Most reports suggest milk has beneficial health effects with very few, if any, adverse effects reported. Organic milk is reported to confer additional benefits due to its lower omega-6–omega-3 ratio, which is due to the difference in feeding practices, with organic cows predominantly pasture fed. Despite the testified animal, host, and environmental benefits, organic milk production is difficult in several regions due to the cost-intensive process and geographical conditions. Finally, we offer perspectives for a better future and highlight knowledge gaps in the organic dairy management system.

## Introduction

1

Milk is among the most versatile and valuable foods in the food industry. In 2018, global milk production reached 843 billion liters, with an estimated value of USD 307 billion and is projected to grow by 22% by 2027 [1]. Approximately 80% of yearly milk production comes from cows, with the rest from other dairy animals like buffaloes, goats, camels, and sheep, according to the Food and Agriculture Organization [2]. Milk is also an essential component of the human diet, consumed by 80% of the world’s population [3]. Milk and dairy products are important sources of macro and micronutrients, including high-quality proteins, fats, calcium, potassium, phosphorus, vitamin D, riboflavin, and vitamin B12 [4,5]. The majority of global milk production is carried out using traditional and intensive systems, collectively referred to as the conventional milk production system [5]. For the purposes of this review, the conventional milk production system, unless otherwise defined, will refer to milk produced from traditional and intensive milk production systems. The intensive use of mechanization, artificial fertilizers, pesticides and antibiotics within the conventional milk production system has raised substantial concerns for the environment, animal welfare, and consumer health [6]. Misuse of these practices can lead to soil, water and air pollution, increased antibiotic resistance spread, loss of biodiversity, and elevated greenhouse gas (GHG) emissions [7]. Moreover, the conventional milk production system, which prioritizes high productivity and profitability, may compromise the nutritional quality of milk and dairy products [8]. The intensification and environmental repercussions of conventional agriculture, coupled with heightened consumer awareness of animal welfare and demand for safer and healthier food options, have prompted a re-evaluation of agricultural policy [9]. This shift has given rise to more environmentally and animal-friendly practices, such as organic agriculture [10,11].

The FAO of the United Nations (UN) broadly defines organic agriculture as “a system that relies on ecosystem management rather than external agricultural inputs” [12]. Organic agricultural production is an alternative farming system rooted in the ethos of sustainable production [13]. The objective is to prioritize the health and welfare of animals, ensuring clean and sanitary conditions for their shelter and nourishment, along with effective waste management [14]. Organic production promotes preventive health measures without the constant use of stimulants or antibiotics, allowing animal access to pastures and providing them with a diet consisting entirely of organic ingredients for optimal nutrition and wellbeing [15]. In contrast to conventional agricultural production, the use of artificial fertilizers, pesticides, herbicides, genetically modified organisms (GMOs), and antibiotics is banned or restricted in organic agricultural production [16,17]. According to the International Foundation for Organic Agriculture (IFOAM) in 2021, organic agriculture was practiced in 191 countries, on more than 76 million hectares of agricultural land by at least 3.7 million farmers, and the size of the organic market reached 125 billion euros [18].

The intake of organic milk, whether in its natural state or as part of dairy products such as pasteurized whole milk, yogurt, cheese, curd, cream cheese and butter continues to grow worldwide [19]. Today, milk and dairy are the most in-demand organic products after organic fruits and vegetables in the organic food market [20]. Organic milk and dairy products, once available only in a few specialized shops, are now widely available to meet increasing consumer demand [21]. In recent years, research on organic milk and dairy products has also increased [22]. Several studies have reported compositional differences between organic and conventional milk [23,24]. For instance, organic milk has consistently been reported to contain significantly higher levels of whey proteins, total polyunsaturated fatty acids (PUFA), n-3 PUFA and vitamin E (α-tocopherol) [25,26]. Organic milk production has also been reported to influence the microbial content of milk [27]. Compositional differences have been linked to conditions associated with organic production such as breed, environment, health status, and feeding regime [25]. The health benefits of milk are associated with the various bioactives mentioned and can be direct, such as contributing to nutrient uptake, bone health and bone density development, and immunomodulatory potential with effects reported starting from as early as childhood [28], while other benefits can be indirect through the gut microbiota by exerting probiotic potential. Organic dairy production is free from antibiotics and chemicals, thus helping in the reduction in antibiotic resistance gene generation and spread. Further, the low ratio of omega 6 to omega 3 fatty acids, and the higher PUFA content are associated with health benefits, though some results are debated [29,30].

Despite the known benefits of organic farming practices, the debate over the advantages of organic milk and dairy products compared to their conventional counterparts persists [31]. Organic farming presents challenges for farmers involving changes in animal husbandry, land, and crop management [21]. Furthermore, the switch is cost intensive, resulting in comparatively low yields and higher estimated product prices [32]. In addition, adhering to strict, mandatory, and country-specific regulations for organic farming and food production, makes the transition a demanding process [33]. Consumers also often express skepticism due to the high prices of organic dairy products and the lack of definitive studies showcasing their benefits [34]. However, as sustainability concerns continue to gain global attention, the organic dairy market is expected to grow [21].

In light of this information, this review aims to explore the latest research on the production and composition of milk produced using organic agricultural practices. We compare organic and conventional milk production systems in terms of practices and impact on the quality of milk. Furthermore, we discuss the human health benefits of organic milk and dairy products and the future challenges and prospects of organic dairy management.

## An Introduction to Organic Milk Production

2

In this section, we provide an introduction to organic milk production and present the regulatory frameworks and principles that guide this farming practice. For this review, organic milk production, unless otherwise defined, will refer to milk from dairy cattle.

### Organic Milk Production Regulations

Organic milk production is permissible exclusively on certified farms, depending on individual countries’ regulations or organizational certifications [35]. Despite sharing fundamental principles, the specifics of organic milk production regulations exhibit notable variations globally, primarily regarding the rules governing pasture access, nutrition, use of antibiotics, and conventional to organic status conversion period, as detailed in Table 1. Subsequently, organic milk products produced in one country may not retain their organic status when exported to another country with distinct legal requirements [21]. Therefore, the diversity in organic regulations may contribute to the variability in organic milk composition between countries [36].

## Milk Production Systems

3

Traditional and intensive milk production systems are collectively referred to as the conventional milk production system. The conventional system dominates milk production practices worldwide, primarily focusing on high productivity [21]. The intensive system is principally performed in developed countries, while milk production in developing countries is carried out in an extensive (traditional) manner [37,38]. The organic milk production sector is experiencing rapid growth, surpassing the expansion rate of other dairy sectors worldwide [21,39]. A summary of the major distinctions between organic and conventional milk production systems is provided in Table 2.

### Conventional Systems

3.1

#### Traditional System

3.1.1

The traditional system relies on pasture as a low-cost primary feed source [40]. Farming practices are primarily determined by the climate and available resources in a given region. Therefore, the traditional system is primarily employed in temperate climates, such as in Ireland and New Zealand, which leads to a seasonal milk supply. Cows are kept outdoors, grazing on pasture during the warmer months of the year. In the winter months, cows are dried off and housed indoors and are fed a diet of primarily pasture-based silage and hay, which is cut and ensiled from surplus pasture earlier in the year. Their feed is typically administered ad libitum (without specialized equipment and calculation of feed rations). When pasture-based feeds alone fail to meet energy requirements of the animal, concentrate supplements are also provided. The ration is not consistent in this feeding system, making it challenging to achieve a balanced diet and can potentially hinder high milk yields [41]. The traditional system offers cows a more natural environment than the intensive system, allowing the expression of normal behaviors [42]. Pasture-based feeding systems have also been demonstrated to beneficially affect the nutritional quality of milk and dairy products [43]. Milk and dairy products obtained from pasture-based diets have larger proportions of beneficial nutrients for human consumption such as PUFA, conjugated linoleic acid (CLA) and n-3 fatty acids than cows fed concentrate diets [44-48]. While the intensive milk production system is supplanting the traditional system, the latter is expected to dominate for the foreseeable future in developing countries [37,38].

#### Intensive System

3.1.2

The intensive system is based on the use of a total mixed ration (TMR) diet administered using a feed truck. The intensive system is primarily performed in countries with climates which make pasture difficult to grow, including the United States, China, and large areas of Europe [49,50]. TMR is a mixture of roughage (grass/maize/corn silage) as well as concentrate feeds supplemented with vitamins and minerals [45]. TMR feeding offers greater opportunities to enhance intake rates and meet nutritional needs more effectively [51]. Furthermore, this system protects animals from extreme weather conditions [52]. The number of dairy cow farms employing the intensive milk production system has grown significantly over the last 20 years [49]. Animal welfare concerns continue to grow regarding indoor TMR feeding systems. These include increased incidences of lameness [53], mastitis [54], mortality [55] and aggressive behavior due to reduced space [42]. Indoor TMR feeding systems also restrict the animals’ ability to express their natural foraging behavior [42]. The development of the partial mixed ration (PMR) feeding system may alleviate some of these concerns. The PMR system combines indoor TMR feeding with the outdoor grazing of fresh pasture by alternating the feeding approaches. PMR feeding has been shown to increase levels of CLA, α-linolenic acid (ALA), vaccenic acid and PUFA significantly compared to TMR feeding [56], in addition to non-significant differences in milk yield and protein content [56].

### Organic System

3.2

Organic milk production is based on maximizing milk production in an environmentally sustainable way, while prioritizing the health and wellbeing of animals. Distinctive variances exist between organic and conventional milk production systems, each presenting its own set of advantages and drawbacks. No single production system can be deemed ideal, as milk production is an ongoing process. The merits of either system hinge on a comprehensive evaluation encompassing longitudinal sustainability, environmental impact, economic factors, and social considerations. There are several fundamental differences between organic and conventional milk production systems [21,23]. In contrast to the conventional systems, the organic system prioritizes the utilization of native cattle breeds [57]. Crops must be fertilized organically, and the use of synthetic and chemical fertilizers, herbicides and pesticides is prohibited, which has been shown to have beneficial effects on soil composition and functionality compared to conventional systems [58,59]. Animals must be provided with organic feed containing ingredients sourced from organic agricultural production, while the inclusion of natural non-agricultural substances is also permitted. For example, vitamins and minerals are sourced from natural substances such as sprouted grains, cod liver oil, and brewer’s yeast. In general, a minimum of 60% of the feed must be sourced from the corresponding farm. Additionally, a minimum of 60% of the dry matter in the feed ration must consist of roughage, green fodder, dried fodder, or silage. During the summer season, cows are provided unrestricted access to pasture vegetation, predominantly comprising low grasses (50%), tall grasses (30%), and legumes (10–20%) [23].

On organic farms, the duration of pasture feeding frequently extends beyond 180 days, whereas on traditional farms, it typically does not exceed 140 days [36]. Organic pastures stand out for their rich sward biodiversity, encompassing various species of grasses, legumes, and herbs. This diversity directly contributes to the nutritional value and quality of fodder and milk produced [43,59]. During autumn and winter, cattle are required to be provided with roughage, comprising silage made from combinations of cereals and legumes or haylage. The inclusion of beets or potatoes in the cattle’s diet is reserved for the winter season [60]. Similar to traditional farming methods, grazing access has been shown to benefit the welfare and behavior of organic cows compared to conventional systems [61,62]. However, organic farms still show a need for improvement, especially regarding animal health [62]. The main problems faced by organic and conventional systems are analogous, with mastitis and lameness identified as particular areas for improvement [63].The use of GMOs, growth stimulants, and synthetic amino acids is also prohibited in organic agriculture [33], while antibiotics may only be utilized in emergencies for veterinary indication. The rise of antimicrobial resistance (AMR) attributed to the excessive use of antibiotics in food-producing animals has become a significant concern [64], especially concerning the risk of developing newly resistant bacteria that could be transmitted from animals to humans [65]. Encouragingly, organic farming has been demonstrated to markedly decrease the occurrence of AMR in dairy cattle compared to conventional farming, globally [66]. Finally, pasture-based systems have been demonstrated to emit less GHG emissions, such as methane (CH_4_) and carbon dioxide (CO_2_), than conventional farms [67,68].

## Impact of Production Systems on Farm Performance and Raw Milk Composition

4

The composition and physical characteristics of milk exhibit considerable variability, influenced by factors such as environment, age, breed, nutrition, parity, stage of lactation, and health [70]. Numerous studies have compared the quantity and quality of raw milk produced using organic or conventional milk production systems [23]. The gross chemical and physical composition of raw milk produced using organic and conventional milk production systems is shown in Table 3.

### Milk Yield

4.1

The primary determinant of the financial success of dairy cow farms is their level of productivity. Organic dairy production has consistently been reported to have reduced milk yields compared to conventional milk production systems [68]. Organic herds generally attain lower milk yields, ranging from 15% to 28% less compared to the yields of a typical conventional cow [23]. Such stark differences in milk yield are typically traced to lower energy intake, through either less concentrated feeding or lower energy content in forages from organic systems [71]. Furthermore, practices such as adjusting grain feeding levels, selecting breeds to enhance cow milk yield, and employing fossil fuel-based fertilizers to boost forage yields are typically linked to the increased yields in conventional systems [36]. Therefore, lower milk yield, and thus lower profitability of organic milk production, could pose obstacles to the continued growth of the organic dairy industry worldwide [72].

### Udder Health and Somatic Cell Count (SCC)

4.2

Somatic cell count (SCC) serves as a crucial diagnostic parameter for assessing the wellbeing of the mammary gland [73]. An SCC surpassing 400,000 cells per milliliter of milk signifies gland inflammation. Inflammation has detrimental consequences on the overall productivity of cows, impacting both the nutritional quality of milk and its suitability for processing [74]. Factors related to management, such as milking hygiene and the cleanliness of cows, play a role in the occurrence of udder infections. These infections can impact both milk yield and composition [75]. Elevated SCC can exhibit a negative correlation with both the yields and percentages of milk protein and fat [76]. Therefore, any conclusions regarding compositional differences between organic and conventionally produced milk should consider udder health as a contributing factor [23]. Conflicting results have been reported regarding increased or decreased SCCs when comparing organic and conventional dairy production systems [23]. Importantly, in most studies which reported significant differences in SCC, the levels were still below 400,000 cells per mL in both conventional and organic milk. At present, the employed farming system appears to have less influence on udder health compared with management factors (e.g., routine teat dipping and seeking veterinary treatment) and animal level variables (e.g., parity, breed) [36,77-79]. Hence, making a generalization about whether organic farmers have a lower tolerance for poor udder health is not feasible due to potential variations in ethical considerations and divergent regulations regarding the use of antibiotics as a treatment option for organic cows among different countries [63]. Therefore, establishing a definitive relationship between SCC and the production system is challenging.

### Microbiological Quality

4.3

The total bacterial count (TBC) is the most widely used measure of microbial quality of raw milk and is measured using several methods including the standard plate count (SPC), plate loop count (PLC), Petrifilm (3M) aerobic count, and flow cytometry methodologies (e.g., Bactoscan, Foss Analytica, Hillerød, Denmark l) [80]. While specific values for SPC vary worldwide, high-quality raw milk should always have a low TBC [81]. Similar to milk SCCs, contradictory results have been reported regarding increased or decreased TBCs when comparing raw milk produced using organic and conventional systems [23]. Differing TBCs across studies have been attributed to management factors and animal-level variables [36,77-79].

The microbiome consists of the microbiota and its “theatre of activity,” encompassing the collective nucleic acids (including viruses and bacteriophages), structural components, and microbial metabolites associated with the microbiota [82]. The existence of a commensal microbiota on the bovine teat canal and teat skin is widely acknowledged [83,84]. However, the demonstration of a commensal bovine milk microbiome has been disputed by methodological issues, sampling difficulty, and a lack of consistency among studies [85,86]. Previous studies have shown that diet has a direct impact on the gut, rumen, and milk microbiota of bovines [87-89]. To our knowledge, only one study has compared the microbiota of dairy cows from conventional and organic farming [27]. This study demonstrated that the microbiome of the cow’s gut and milk was significantly different between agricultural management systems, while no differences were found in the microbial communities of soil and silage [27]. Milk samples from organic farms were significantly associated with the family *Rhodobacteraceae* and elevated levels of *Ruminococcaceae*. Furthermore, there was a notable association of the fungi Dothideomycetes, Tremellomycetes, and Pleosporales with milk samples from organic farms. Fungi within these classes are commonly associated with plant pathogens that thrive on wood debris or decaying leaves. Nevertheless, their presence has been reported in the dairy farm environment [90] and on shelves used for ripening cheese [91].

### Mastitis

4.4

Mastitis stands out as the most widespread and economically impactful disease in dairy cattle globally, primarily attributed to diminished milk production, discarded milk, premature culling, and associated treatment expenses [92]. Bovine mastitis is a polymicrobial disease with the principal etiological agents being *Staphylococcus aureus, Streptococcus dysgalactiae*, and *Streptococcus uberis* [93]. Although treatment with antibiotics is the last resort for organic farmers, their usage is permitted under the prescription of a veterinarian [78]. Antibiotics are currently the preferred treatment for mastitis control on both organic and conventional farms [94]. While the epidemiology of mastitis on organic farms has not been extensively studied, available reports suggest that organic farms have an elevated prevalence of *Staphylococcus aureus* compared with conventional dairy farms [95-98]. The incidence of clinical mastitis on organic dairy farms has been reported to be lower than on conventional farms [99-101]. Additionally, no differences have been found in the incidence of subclinical mastitis [102] or individual SCC [103] on organic versus conventional farms. Such reports suggest that there may be differences in mastitis epidemiology between conventional and organic dairy farms. Future studies are needed to assess the antimicrobial resistance profiles and ubiquity of antibiotic-resistant bacteria, such as Methicillin-resistant *Staphylococcus aureus* (MRSA) in mastitic milk from conventional and organic dairy farms.

### Volatile Organic Compounds

4.5

Milk contains low concentrations of volatile organic compounds (VOCs) which are influenced by several variables, such as environment, breed, and lactation stage [104]. VOCs have consistently been associated with the sensory profiles of milk products, especially odors and flavors [105]. VOCs emanate in milk via metabolic processes of the cow (e.g., rumen gases, blood, etc.) or can be infused into milk through animal feed, which influences the flavor of dairy products [106-108]. There have been conflicting reports regarding variation in the VOC composition of milk produced using the organic and intensive production systems have been reported [36,109-111]. VOCs markers, such as terpenes, warrant further exploration for their potential to authenticate dairy products [26,70,112].

Animal feeding systems (pasture or TMR) have been shown by numerous studies to alter the sensory characteristics of milk and dairy products [113]. Some studies have found little differences in the flavor and texture of milk and dairy products produced using organic and conventional systems [36,109,114]. Studies have also indicated that raw organic milk was creamier and tended to have greater ‘hay’ and ‘grass’ flavor notes than conventional milk [115]. Irrespective of production system, a stronger odor of milk, butter, and cheese (more intense ‘animal’ notes) has repeatedly been reported when cows are pasture-fed vs. fed on conserved forages [46,116].

### Protein

4.6

The total protein and casein content of organic and conventionally produced milk is typically reported to not differ significantly [117-119]. Whey proteins, while making up only 20–25% of the total protein, constitute a crucial group of milk proteins (the remaining 75–80% is casein). Albumins, i.e., α-lactalbumin (α-LA), β-lactoglobulin (β-LG), and bovine serum albumin (BSA), make up approximately 75% of whey proteins. Other minor whey proteins include bacteriostatic substances, i.e., immunoglobulins, lactoferrin, lactoperoxidase, and lysozyme, which constitute 1–2% of total milk proteins. These proteins exhibit diverse positive effects on the human body, encompassing antimicrobial (antiviral and antibacterial), anticancer, immunomodulatory, and antioxidant properties. Whey proteins serve as an excellent source of energy, essential amino acids, and peptides [120]. The concentration of whey proteins and albumins in organic and conventionally produced milk is largely similar in studies to date [117,121,122]. Recent studies have indicated that concentrations of lactoferrin and lysozyme are significantly higher in milk on organic farms than on conventional farms [121,123]. Lysozyme is an antimicrobial enzyme that induces cell lysis by hydrolyzing the peptidoglycan layer of both gram-positive and gram-negative bacterial cells. When ingested, lactoferrin induces various beneficial biological effects, such as enhancing iron absorption, modulating the immune system, boosting the antimicrobial activity of lysozyme, and promoting the growth of epithelial cells and fibroblasts [124].

### Vitamins

4.7

Vitamin A (retinol) serves as the precursor to a group of compounds known as retinoids, which exhibit the biological activity associated with vitamin A. Vitamin A encompasses a group of analogous fat-soluble vitamins that play ubiquitous roles in the human body, such as enhancing vision, cell differentiation, embryogenesis, reproduction, growth and immune system functioning [125]. In general, foods of animal origin provide preformed vitamin A as retinyl esters while plant-derived foods provide precursors of vitamin A, i.e., carotenoids. Only carotenoids with a β-ionone ring (e.g., β-carotene) can function as precursors of vitamin A [126]. In cow’s milk, vitamin A is typically found in the forms of retinol or β-carotene [126]. The concentration of vitamin A and carotenoids in milk is significantly influenced by the carotenoid content of the animal’s diet. Milk from animals fed on pasture generally contains higher levels of carotenes compared to milk from animals fed on concentrate feeds [117]. Vitamin E constitutes a group of fat-soluble molecules which primarily act as antioxidants in cell membranes where the primary function is to prevent oxidative damage by trapping reactive oxyradicals [127]. Vitamin E is also essential for body functions in both bovines and humans such as growth, reproduction, immunity prevention, and protection of tissues [128]. β-carotene and Vitamin E concentrations differ significantly in raw milk depending on the diet [25]. The dairy industry is interested in a high content of vitamin E and β-carotene, as they can prevent the spontaneous oxidation of milk and fatty acids [129]. Vitamin D_3_ plays a crucial role in the metabolism of calcium and phosphorus, contributing to the proper mineralization of bones and teeth. Additionally, it exhibits immunomodulatory and anti-cancer properties. In the case of animals spending time at pasture, ultraviolet (UV) rays from sunlight induce the synthesis of vitamin D_3_ from 7-dehydrosterol present in the skin. Therefore, milk from cows that spend more time outdoors at pasture is expected to be a more valuable source of this vitamin. Numerous studies have reported higher vitamin D_3_ levels in milk from cows of organic and traditional production systems compared to intensive systems [117,122,130].

### Carbohydrates

4.8

Lactose is the main carbohydrate in milk and is generally reported to not significantly differ in the feeding system [23]. Oligosaccharides are the third most abundant solid component found in milk, after lactose and lipids [131]. These structurally and biologically diverse molecules, despite being resistant to human digestive enzymes, are linked to numerous beneficial functions [132]. Organic and conventional pasture-based farming systems have been demonstrated to not significantly influence oligosaccharide abundance [110]. However, levels of specific oligosaccharides were increased in organic milk irrespective of sampling date or farm set [110], specifically, trisaccharides with three hexose units (3 Hex), trisaccharides with three hexose units and one N-acetylneuraminic acid unit (3 Hex, 1 NeuAc), tetrasaccharides with four hexose units and one N-acetylhexosamine unit (4 Hex, 1 HexNAc), and trisaccharides with three hexose units and two N-acetylhexosamine units (3 Hex, 2 HexNAc) [110].

### Fats

4.9

The total fat content of organic and conventionally produced milk is typically reported to not differ significantly. Milk fat consists of over 400 different fatty acids. The predominant fatty acids in milk are saturated fatty acids (SFA), with unsaturated fatty acids, including monounsaturated fatty acids (MUFA) and polyunsaturated fatty acids (PUFA), following. Nevertheless, recent scientific advancements have suggested that trans fatty acids and certain saturated fatty acids in milk may have beneficial effects [113,133,134]. The concentrations of individual fatty acids in milk fat are affected by factors such as cow breed, stage of lactation, genetics, and diet [113,135]. The composition and quantity of fatty acids in milk are primarily dictated by the feeding system [25,136].

Fresh herbs and grasses in the cow’s diet contribute a significantly higher quantity of unsaturated fatty acids, whereas maize silage has a greater concentration of linoleic acid [113]. The TMR feeding system markedly diminishes the fat and fatty acid content in milk, attributed to the insufficient dietary fiber and elevated starch levels in the diet [137,138]. Organic milk has consistently been shown to contain a more favorable fatty acid profile than conventional milk [139], containing more PUFAs, including omega-6 and omega-3 [24,139-141] and a lower ratio of omega-6 to omega-3 fatty acids, which is beneficial for human health [139,142-146]. The omega-6 to omega-3 fatty acid ratio in bovine milk essentially characterizes the concentrations of linoleic acid versus α-linolenic acid, as they represent the most abundant omega-6 and omega-3 fatty acids, respectively.

Forage is abundant in α-linolenic acid, while cereals such as barley, maize, oats, and soybean contain higher quantities of linoleic acid [147]. A lower omega-6 to omega-3 fatty acid ratio is therefore suggestive of a forage-based diet [113]. Organic milk has also been shown to contain higher Conjugated Linoleic Acid (CLA) content than conventional milk [148,149]. The consumption of milk and dairy products rich in CLA is linked to beneficial effects on human health, including improved brain function, antiatherogenic effects, and lower levels of blood lipids [150,151]. CLA also demonstrates anti-carcinogenic, immunostimulatory, and weight-reducing properties [151].

### Minerals and Heavy Metals

4.10

The mineral content of milk is influenced by a variety of factors including animal diet, genetics, breed, feeding system, and the surrounding environment [152]. The concentration of minerals in milk is primarily contingent on their levels in fodder [153,154]. The mineral content of forage is determined by the mineral content of soil and pasture, which is influenced by fertilizers, the amount of sewage sludge generated, soil type, or the proximity of mining and industrial areas [36]. In conventional farming, soil fertility can be increased by using mineral fertilizers enriched with selected microelements [155]. Cow diets are also supplemented with mineral mixtures to increase the mineral content of milk produced [23]. Both of these methods are restricted in organic farming; therefore, on-farm fodder is the main source of minerals [23]. Green forage from legume plants offers substantial amounts of calcium and magnesium. Cereal grains provide phosphorus, wheat bran serves as a source of magnesium, and green forage contains smaller amounts of sodium [23]. Organic milk has generally been reported to contain a marginally lower mineral content than conventionally produced milk, with the difference primarily attributed to management practices [24,25,156,157]. These practices include selenium supplementation to improve reproductive performance, iodine-containing teat dipping as a disinfectant after milking, and mineral supplementation [24,25,156,157]. Toxic elements, including heavy metals, such as lead, chromium, mercury, and cadmium may also be present in milk and dairy products [158]. Such heavy metals are non-essential elements, have no biological role in mammals, and can cause toxic effects even at very low concentrations [159]. The main source of heavy metals in agricultural systems is fertilizer [160]. Numerous studies have reported significantly higher levels of heavy metals in conventionally produced milk [119,161,162].

## Perceived Health Benefits of Organic and Conventional Milk

5

Milk and dairy products provide several health benefits beginning from the early stages of life. Recommendations based on guidelines for several countries across the globe include milk and other dairy products as part of daily healthy eating [166]. Research over the past two decades has delineated the associations between milk consumption and health benefits. Some of the highest increases in the numbers of diseases worldwide are seen concerning obesity, type 2 diabetes, and cancer. A recent meta-analysis reported that children consuming higher dairy intake had lower incidence of overweight compared to those having lower dairy intake [167]. However, another meta-analysis failed to show any association in children, though a slight positive association with a protective effect of dairy consumption was reported in adolescence [167]. Other comprehensive short-term studies even report the role of dairy products in facilitating weight loss in an energy-restricted diet, though long-term studies fail to provide convincing results for the same [168,169]. Similarly, other meta-analyses reported only a slight positive effect or no effect of dairy on diabetes [170,171] and mixed or no association with risk of cardiovascular diseases [170,172]. Another meta-analysis reported a positive role of dairy (particularly yogurt) in preventing the risk of type 2 diabetes; however, no association with milk and a negative association with cheese consumption was reported [173,174]. Furthermore, as observed from several meta-analyses, total dairy, full-fat dairy, low-fat dairy, milk, cheese, and yogurt consumption have no association with the risk of coronary heart disease [170,175]; though, controversial results with slightly positive effects of dairy consumption on preventing risk of cardiovascular disease were reported based on prospective cohort studies [176,177]. Variations in results based on different dairy products can be because of their potential impact on the host microbiome, the variations in practices to prepare them (such as fermented vs. not fermented), and the levels of nutrients in different dairy products. Inconsistent results observed can also be attributed to the varying nutrient content of the milk due to the varying laws and practices used by farms around the world.

Similarly, due to the high calcium and magnesium levels in milk, several studies have associated milk intake in early life with a lower risk of osteoporosis and fracture incidence [178,179]. Another meta-analysis reported that high dietary calcium through dairy with or without vitamin D supplementation increases body and lumbar bone mineral content, though this effect was only seen in children in the low baseline group as opposed to the high baseline dairy intake group [180]. Similar results of the possible effects of calcium along with vitamin D supplementation in reducing the risk of osteoporosis and bone fractures are reported in adults [28]. This points towards the need for further studies to understand the optimum levels of dairy and calcium intake to support bone mineral content and density in children. Furthermore, calcium, magnesium, and other nutrients from milk have similar benefits in adults and contribute to bone health and maintain bone structure [181,182].

In population studies, the relationship between dairy consumption and cancer risk has yielded mixed results, with limited and often inconclusive data. The bioactive compounds in dairy could have both positive (linked to calcium, lactoferrin, and fermentation products) and negative (linked to insulin-like growth factor I (IGF-1)) effects on cancer development. The World Cancer Research Fund (WCRF) continually reviews evidence on diet and cancer prevention, and some findings suggest that dairy, particularly milk and calcium, may reduce the risk of colorectal cancer [183]. Similar results were reported by other meta-analyses [180,184]. However, the evidence regarding breast cancer is inconclusive [183], although some studies suggest a potential protective effect of dairy intake, especially yogurt and low-fat dairy [185,186]. While according to the WCRF, 2014 and 2015 reports, and other observational studies, mixed results with limited evidence have been reported for associations between dairy and risk of prostrate and bladder cancer [183,187].

As reviewed earlier in Section 4, though the nutritional composition of conventional and organic milk is very similar, studies have reported differences in the levels of these nutrients in the two milk types. These differences can lead to enhanced health benefits as perceived and claimed by organic milk. It is important to consider these nutrients in the recommended daily reference intake, so as to understand the benefits, if any. Studies regarding the fatty acid composition are fairly consistent due to the direct effect of diet on milk fatty acid composition; however, protein and carbohydrate compositional results vary between studies. For instance, as mentioned above, it is now confirmed by several studies that organic milk contains higher n-3 PUFAs, CLA, and a lower omega-6 to omega-3 ratio than conventional milk [24,188]. The meta-analysis by Średnicka-Tober et al. also reported higher levels of α-tocopherol, β-carotene, lutein, and vitamin E in organic than conventional milk—an imbalance in omega 6–omega 3 ratio is associated with cardiovascular disease risk, cancer, and hypertension and disease pathogenesis [189]. As reviewed by Givens and Lovegrove, the differences in fatty acids between organic and conventional systems in the context of overall diets are important but are minimal, thus further studies with larger sample sizes are needed to underline the association between organic milk and health benefits [24]. Similarly, Średnicka-Tober et al. also report lower levels of iodine and higher levels of iron in organic milk compared to conventional milk [24]. However, milk is not the primary source of iodine or iron for humans and an otherwise balanced diet must be used to maintain levels of these nutrients. If milk is the source of iron for individuals, then those consuming organic milk must consume the appropriate supplements to avoid deficiency. These differences in nutrients are predicted to be observed under the circumstances of the switch to organic dairy and can impact health [30]. Some studies suggest the positive associations of organic dairy consumption with a lower risk of eczema in children [190] and a higher prevalence of hypospadias in the male offspring of mothers consuming conventional over organic dairy products [191].

A key characteristic of organic milk farming is avoiding the use of antibiotics and pesticides, as this can help enhance the efficacy of existing antibiotics in animals and humans. Even though this might not affect the nutritional composition of milk, it must be noted that this will lead to a reduction in the generation of new antibiotic resistance genes (ARGs) and the selection of antibiotic-resistant bacteria, thus lowering the chances of the spread of ARGs to the calf and the environment. Furthermore, the limited or prohibited use of antimicrobials and chemicals can positively impact the microbial quality of organic dairy with lower numbers of antibiotic-resistant bacteria, but safety concerns are prevalent, such as the risk of foodborne illness [192].

## Global Market for Organic Milk Products

6

In the past decade, increasing awareness of self-health and the environment has given much importance to the holistic approach to organic food production. Consumption of organic products by consumers relies partly on their behavior with optimistic consumers more inclined towards organic products than pessimistic consumers, with environmental concerns driving higher consumption by pessimistic consumers [193]. Overall, the health benefits, sensory appeal, and quality of organic food products are some of the prominent factors along with environmental concerns for consumers [194]. Owing to these concerns, organic dairy products are no longer confined to first-world countries but form a big market across populations throughout the world. The global organic dairy market is estimated to be worth about $54 billion US dollars by 2030, nearly more than double the $24 billion US dollars in 2021 [195]. Asia is one of the biggest contenders in organic dairy production, followed by North America.

Organic dairy certification varies between countries; however, some standards as laid out by the United States Department of Agriculture (USDA) under the Organic Foods Production Act of 1990 and the National Organic Program (NOP), the European Union Regulation (2018/848) (https://eur-lex.europa.eu/eli/reg/2018/848/oj, accessed on 14 November 2024), Draft Guidelines of Codex/WHO/FAO and the International Federation of Organic Agricultural Movements (IFOAM) are similar and accepted on a large scale. In the East, Southeast, and South Asia, the Asia Regional Organic Standard (AROS) sets the regulation (https://www.fao.org/family-farming/detail/en/c/282204/, accessed on 8 November 2024). However, to be certified as organic, each farm must comply with the regulations governing that area. These regulations can vary widely, such as the use of antimicrobials for the treatment of mastitis in organic dairy farms is strictly prohibited in the US but permitted under veterinary recommendation in the European Union [196].

Organic dairy is the second most consumed category after fresh fruits and vegetables in the US, with retail sales totaling approximately $6 billion in 2020. In the US, an increase in certified organic dairy cows from 2000 to 360, 000 has been observed from 1990 to 2019. The increase in organic fluid milk sales from 1.92% to 5.5% of the total sales was observed from 2009 to 2021; though the increase almost plateaued by 2014 in the US [197]. By 2019, European countries including Austria, Denmark, Germany, and France had the highest numbers of organic dairy cows in the total dairy herd (Eurostat, ING research). In Europe, Germany, France, Denmark and Austria also produced the highest volume of organic milk in 2017 [198]. However, an increase in the number of organic dairy cows from 2012 to 2019 was only about 2%, which is forecasted to be much higher by 2030 [198]. The total increase in organic milk production from 2007 to 2015 also doubled in Europe, just like in the US [199]. By 2019, organic milk production in the EU represented 3.5% of total EU milk, which is also expected to be around 8% by 2031 [200].

India is the largest milk producer in the world, with an increasing demand for organic dairy. The geographical and climatic conditions in some regions, and the disease-resistant native breeds in particular provide additional benefits and are well suited for organic farming. However, cost concerns and limited knowledge of organic farming in small farms leave a large gap and a plethora of opportunities to maximize the capacity of dairy farming in India [201,202].

Despite New Zealand and Australia being leading milk producers and exporters in the world, their contributions to global organic milk were ranked 15th and 20th, respectively, in 2017 [203]. A decreasing trend in milk production has been reported in Australia over the last decade. Though the majority of dairies in Australia are in coastal areas, allowing access to fresh grazed grass, a shift to concentrate feeding has been observed due to cost and climatic conditions [43]. This could be because of the weather conditions restricting the production system.

The Japanese Agricultural Standards (JAS) for Organic Agricultural Products was established in 2000 with further development in 2005, with the addition of the JAS for Organic Livestock Products. Organic farming is faced with several difficulties in Japan due to the nature of the climate and crops prone to pests making the use of pesticides unavoidable to an extent. However, it is strictly regulated by the JAS in Japan and due to the environment-friendly approach involved, it is projected to increase [204].

## Future Challenges and Perspectives

7

The organic dairy production system is associated with several perceived health benefits for consumers, and most importantly with animal health and welfare. However, it is crucial to acknowledge that the level of animal welfare can differ greatly within each production system. At present, there is no substantial evidence to support the claims that animal welfare is better in the organic or conventional system [63]. The prohibitive/restricted use of antimicrobials has invoked farmers to use various antibiotic alternatives such as aloe vera and whey-based products for disease treatment [96], and has created opportunities for several antimicrobial alternatives to be used in the dairy industry. However, the transition from conventional to organic farming is expensive and requires changes to animal husbandry, farm practices and land management. Along with these changes, certification and compliance work is very challenging accompanied by higher costs for animal maintenance [205]. Thus, incentives and support from the government are required, especially in developing countries, to provide a boost for farmers to switch to organic farming. This will help meet the challenge of demand and supply of organic dairy, which might rise with the growing demand for organic food products. Further, studies are also needed to evaluate the nutritional benefits of organic milk with the recommended daily dietary intake. Along with this, appropriate education for consumers is necessary, allowing a well-informed decision among consumers.

The benefits of organic dairy are suggested to be associated with the feed type; however, other factors including farm management and breed type are also variables in the composition of organic and conventional milk [110]. Moreover, the claimed differences in the composition of organic and conventional milk are sometimes associated with differences in the abundance of bacteria such as lactic acid bacteria, which is reported to be higher in organic dairy favored by higher concentrations of peptides and long-chain PUFAs [206]. This points towards the need for further studies to understand the additional probiotic, growth-promoting effects, and microbiological safety of organic dairy products.

## Figures and Tables

**Table 1 T1:** Country-specific organic dairy farming regulations regarding pasture access, forage feeding, antibiotic usage, and conventional to organic status conversion period. Adapted from [36].

Country	Pasture Access	Nutrition	Antibiotics Use	Organic Conversion Period	Regulation
European Union	Year-round, weather permitting	≥60% of daily dry matter intake must consist of roughage, fresh or dried fodder, or silage.	Permitted under veterinary recommendation. ≥2 day milk withdrawal. ≥3 treatments or ≥1 treatment (if productive lifecycle is <1 y) will cause animal to lose its organic status.	Land conversion period of 24-months. Animals must be under organic management ≥6 months.	Regulation (EU) 2018/848 of the European Parliament and of the Council.
United States	≥120 days annually	≥30% of daily dry matter intake must come from pasture during grazing season.	Prohibited. Usage will cause animal to lose its organic status.	Animals must be under organic management ≥12 months.	Organic foods production act provisions 2023.
Canada	≥120 days annually	≥30% of daily dry matter intake must come from pasture during grazing season. 60% of dry matter intake consists of hay, fresh/dried fodder, or silage.	Permitted under veterinary recommendation. ≥30 day milk withdrawal. ≥2 treatments, 12 month transition period before regaining organic status.	Animals must be under organic management ≥12 months.	Organic Production Systems General Principles and Management Standards 2021.
Japan	≥2 days per week, year-round	≥50% of daily dry matter intake must consist of roughage, fresh or dried fodder, or silage.	Permitted under veterinary recommendation.	Animals must be under organic management ≥6 months.	Japanese Agricultural Standard for Organic Livestock Products, 2018.
New Zealand	≥150 days annually	≥50% of daily dry matter intake must consist of roughage, fresh or dried fodder, or silage.	Prohibited. Usage will cause animal to lose its organic status.	Animals must be under organic management ≥12 months.	AsureQuality Organic Standard For Primary Producers, 2018.
Australia	Year-round, weather permitting	100% of daily dry matter intake must be sourced from organic or bio-dynamic feed.	Permitted under veterinary recommendation. 180 day transition period before regaining organic status.	Animals must be under organic management ≥6 months.	National Standard for Organic and Bio-Dynamic Produce, 2022.
China	Year-round, weather permitting	≥60% of daily dry matter intake must consist of roughage, fresh or dried fodder, or silage.	Permitted under veterinary recommendation.	Animals must be under organic management ≥6 months.	China Organic Standard GB/T 19630-2019.
India	Year-round, weather permitting	≥85% of daily dry matter intake must be sourced from organic feed	Permitted under veterinary recommendation.	Land conversion period of 24 months. Animals must be under organic management ≥6 months.	Agricultural and Processed Food Products Export Development Authority (APEDA) 2018.

**Table 2 T2:** Management Practices of Organic and Conventional Milk Production Systems. Adapted from [69].

	Milk Production System	
Management Practice	Organic	Conventional
Pasture access	Required	Not required
Nutrition	All feed must be certified organic	Concentrate feed
Antibiotics use	In emergencies, for veterinary indication	Allowed, for veterinary indication
Parasiticide use	In emergencies, for veterinary indication	Allowed, for veterinary indication
Growth Hormone use	Prohibited	Allowed, for veterinary indication
Weed Management	Crop rotation, hand weeding, mulches	Chemical Herbicides
Pest Management	Crop rotation, Companion Planting, trap crops, promotion of beneficial insects and natural predators	Chemical Pesticides
Green House Gas Emissions	Lower per unit of area	Higher per unit of area
Fertilizers	Organic fertilizers only	High dependence on synthetic NPK fertilizers
Genetically Modified Organisms	Prohibited	Allowed
Synthetic food Additives	Prohibited	Allowed
Milk Yields	Lower on average	Higher on average
Shelf Life	Higher on average	Lower on average
Product Price	Higher on average	Lower on average
Soil Impact	Reduced soil loss, increased organic matter, water-holding capacity and microbial diversity	Increased soil loss and erosion, lower water holding capacity, lower carbon storage and microbial diversity
Water Consumption	Lower	Higher
Energy Usage	Low intensity of energy use (higher energy efficiency)	High intensity of non-renewable energy use (agrochemicals, machinery, water pumping etc.)
Impact on Landscape	Larger floral and faunal biodiversity. Diverse agricultural landscapes	Loss of biodiversity in agricultural landscapes, Unified agricultural landscapes (monocultures)

**Table 3 T3:** Concentrations of select macronutrients, micronutrients, and general antimicrobial peptides present in raw milk produced using organic, traditional and intensive systems. Traditional milk refers to milk produced using the traditional milk production system. Intensive milk refers to milk produced using the intensive milk production system.

	Organic System	Conventional Systems	
Proteins	Organic Milk	Traditional Milk	Intensive Milk
Total Protein (%)	3.1−3.26	3.1−3.24	3.48
Casein (%)	2.54	2.52	2.78
Whey protein (%)	0.72−0.84	0.72−0.84	0.70−0.82
β-Lactoglobulin (g/L)	3.32−3.35	3.26−3.58	3.01−3.28
α-Lactalbumin (g/L)	1.07−1.19	1.05−1.21	0.98−1.14
Bovine serum albumin (g/L)	0.43	0.44	0.41−0.49
Lactoferrin (mg/L)	123.8−125.9	109.80−130.62	94.01−121.23
Lysozyme (μg/L)	11.14	9.92−10.71	6.90−12.13
Vitamins	Organic Milk	Traditional Milk	Intensive Milk
Vitamin A (retinol) (mg/L)	0.468−0.800	0.410−0.556	0.347−0.465
β-carotene (mg/L)	0.195−0.580	0.231−0.252	0.175−0.190
Vitamin E (a-tocopherol) (mg/L)	1.358−2.655	1.656−1.953	1.075−1.302
Vitamin D_3_ (cholecalciferol) (μg/L)	0.461−0.768	0.610−1.212	0.589−0.700
Carbohydrates	Organic Milk	Traditional Milk	Intensive Milk
Lactose (%)	4.80−5	4.7−5	nd
3 Hex (Trisa) (*m/z*)	60.82−61.11	51.37−55.86	nd
3 Hex, 1 NeuAc (*m*/*z*)	11.83−14.60	9.24−12.42	nd
4 Hex, 1 HexNAc (*m*/*z*)	0.87−0.93	0.63−0.69	nd
3 Hex, 2 HexNAc (*m*/*z*)	0.31−0.33	0.25	nd
Fat	Organic Milk	Traditional Milk	Intensive Milk
Fat (%)	3.7−4	3.8−4	3.8−4
SFAs (g/100 g)	66.28	59.03−64.74	67.69−71.41
MUFAs (g/100 g)	26.11−34.07	30.33−32.16	21.87−28.15
Oleic acid (c9 C18:1)	20	16.10−22.66	16.16−17.20
Vaccenic acid (t11 C18:1) (g/100 g)	1.22−2.00	1.18−7.00	0.80−2.00
PUFAs (g/100 g)	3.85−5.36	3.69−5.32	1.65−3.77
Eicosapentaenoic acid, EPA (C20:5 n-3) (g/100 g)	0.05	0.08	0.05
Conjugated linoleic acid, CLA (cis9 trans11) (g/100 g)	0.83−1.53	0.54−0.93	0.42−1.19
Linoleic acid, LA (C18:2 n-6) (g/100 g)	0.59−2.08	1.17−2.18	1.4−2.39
α-linolenic acid, ALA (C18:3 n-3) (g/100 g)	0.44−1.05	0.49−1.25	0.39−0.42
γ-linolenic acid, GLA (C18:3 n-6) (g/100 g)	0.11	0.13	0.12
Proportion 18:3n3: 18:3n6	1.35	0.60−2.77	1.26
Minerals and Heavy Metals	Organic Milk	Conventional Milk	
Calcium (mg/L)	971.33−1161	1170−1417.76	
Iron (mg/L)	0.26−0.67	0.26−0.47	
Manganese (mg/L)	0.023−0.047	0.022−0.139	
Copper (mg/L)	0.023−0.084	0.038−0.161	
Iodine(mg/L)	0.013−0.283	0.071−6.540	
Aluminium (mg/L)	0.76	0.63	
Potassium (mg/L)	1509−1896.92	1514−1844.37	
Sodium (mg/L)	366.59	476.35	
Magnesium (mg/L)	86.21	113.87−118.50	
Zinc (mg/L)	2.86−3.96	2.96−4.39	
Selenium (mg/L)	0.002−0.020	0.008−0.040	
Cobalt (mg/L)	0.001	0.001	
Strontium (mg/L)	0.166	0.202	

Ranges are shown where available. Values for the traditional system and intensive system are shown where available. Values obtained from [23,110,142,144,163-165]. Abbreviations: Trisa, Trisaccharides; Hex, glucose or galactose; HexNAc, N-acetylglucosamine or N-acetylgalactosamine; NeuAc, N-acetylneuraminic acid (sialic acid); (*m*/*z*), mass divided by charge number; MUFAs, monounsaturated fatty acids; PUFAs, polyunsaturated fatty acids; SFAs, saturated fatty acids; nd, no data.

## Data Availability

No new data were created for the production of this manuscript. All of the data here discussed and presented are available in the relative references here cited and listed.

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
