# Peer review of "Production, Composition and Nutritional Properties of Organic Milk: A Critical Review"

_foods, 2024, doi:10.3390/foods13040550_

Round 1
Reviewer 1 Report
Comments and Suggestions for Authors
Recommendations:
It is suggested to review the manuscript and consider that there are evident biases in the comparisons, of which a tendency can be observed towards broadly describing the organic milk production system, which we hope the authors can improve and avoid for a future review, among which the following stand out:
1) Regarding the development of the manuscript, there is an abundance of details comparing the organic milk production system with the conventional system, but there is no mention that defines the latter system.
2) Clearly in the conventional system there is a universe of extensive and intensive productive systems that are only mentioned in Table 3 and each of them has different productive perspectives. To clarify this issue, it is suggested to search for bibliographic information. To help, it is suggested to see WOAH, 2022. CHAPTER 7.11. ANIMAL WELFARE AND DAIRY CATTLE PRODUCTION SYSTEMS.
https://www.woah.org/fileadmin/Home/eng/Health_standards/tahc/current/chapitre_aw_dairy_cattle.pdf
Phillips C.J.C., 2018. Principles of Cattle Production, 3rd Edition, CABI. 271 pages. Pondicherry, India
3) In the comments there are suggestions related to central aspects when considering animal welfare, also with considerations aimed at disqualifying commercial production systems by overvaluing the animal welfare aspects of the organic system. It is suggested to review the existing scientific information and evaluate it with evidence in its proper measure.
4) Milk production systems about animal welfare conditions are always a continuum from very good to bad and you can find as much intermediate data as you want. When you want to compare the different production systems about animal welfare, all systems have advantages and disadvantages. In the following graph, you can see the comparison of the criteria and their relationship with the pasture and intensive systems.
5) The manuscript describes interesting aspects carried out through a review of the organic milk production system, the title refers to a critical review, but nothing is mentioned about a comparative review with what the authors called the “conventional system”, perhaps You could consider improving it by precisely making a path towards criticism of the system itself and if you want to compare it as mentioned in WOAH, 2022: a) housed: systems where cattle are kept on a surface formed, indoors or outdoors; b) Pastured: These are systems where cattle live outdoors; c) Combination systems: These are systems where cattle are managed in any combination of housed and pasture production systems
It is suggested to review and adjust the following:
Line 28 -29: It is recommended to enhance this expression, for example: Finally, we offer perspectives for a better future and highlight knowledge gaps in the organic dairy management system.
Line 37 - 40: It is recommended to enhance this expression, for example: Approximately 80% of yearly milk production comes from cows, with the rest from other dairy animals like buffaloes, goats, camels, and sheep, according to the Food and Agriculture Organization (FAO, 2016). Review, cite, and create bibliography with assistance from internet search on 01/15/2024: https://www.fao.org/3/cb2992en/cb2992en.pdf
Line 17: Review the terminology, use WOAH, 2022 or another accepted by the science of animal production.
Line 21: Same as the previous one.
Line 69: … to the …
Line 71: .... Agriculture, 2023). ...
Line 74: … yoghurt …
Line 93: … helping …
Line 109: … , this review aims to explore ...
Line 125: … the heterogeneity …
Line 134: … focusing …
Line 136: Here it is backwards, extensive (traditional). The technical term used by science is extensive, the vulgar term is traditional.
Line 165: … based on the use …, … mixed of ration …
Line 171: … opportunities …
Line 181: … feeding the system …
Line 183: … to increase levels of CLA significantly …
Line 188 -189: It is a controversial topic. It is important to consider that animal welfare can vary greatly in each production system. Review and improve this sentence "Organic milk production is based on maximizing milk production in an environmentally sustainable way, while prioritizing the health and well-being of animals." Although it refers to animal welfare in organic production, it must be taken into account that there is no ideal production system. Thinking that well-being is a continuum affects production, there are sustainability, environmental, economic and social aspects that can be affected in all systems, please review, they are not exclusive to the organic system.
Line 189 – 190: … There are several fundamental …
Line 192: …. the utilization ….
Line 213: … in emergencies for veterinary indication.
Line 215: … especially concerning the risk …
Table 2:
Antibiotics use |
In emergencies, for veterinary indication |
Allowed, for veterinary indication |
Parasiticide use |
In emergencies, for veterinary indication |
Allowed, for veterinary indication |
Growth Hormone use |
Prohibited |
Allowed, for veterinary indication |
Animal Welfare Standards (1) |
High animal welfare standards |
Significantly lower animal welfare standards |
(1) This is a contentious issue. It's crucial to note that the level of animal welfare can differ greatly within each production system, and it's essential to implement good practices in nutritional, environmental, behavioral, and preventive management of the physical and mental/emotional health of cattle in both systems. The decision between organic and conventional systems often hinges on personal values, market demand, and local regulations. It is recommended to review and enhance this aspect. It is important to consider the specific needs and preferences of consumers and to keep abreast of any changes in market trends and regulations. Both organic and conventional production systems have their own set of advantages and challenges, and it is crucial to weigh these factors carefully when making a decision. Additionally, ongoing research and innovation in the field of animal welfare and sustainable agriculture may provide new insights and opportunities for improvement in both systems.
Aim (2) |
Maximizing environmentally stable production and prioritizes the health and well-being of animals |
Maximizing the economical effectiveness of the production |
(2) This is another controversial topic. Organic systems often have higher production costs due to the need to meet organic standards, which may include specific agricultural practices, organic feeding requirements, among other aspects. Organic milk is often sold at a higher price in the market due to the increasing demand for organic products and the additional costs associated with organic production. In some regions, organic farmers may have access to specific government subsidies and supports to encourage sustainable and organic farming practices. It is sometimes associated with lower yields due to restrictions on the use of certain agricultural inputs. However, this may depend on agricultural management and adaptation to organic practices. In addition, it can open opportunities in specialized markets and niches of consumers willing to pay a higher price for organic and sustainable products. Organic certification may involve additional costs to meet the standards and audit processes necessary to maintain organic status.
While conventional systems may have lower production costs in terms of organic requirements and may benefit from economies of scale in purchasing conventional agricultural inputs. Milk may have a lower retail price compared to organic milk, but demand is usually constant due to its widespread availability and they may also benefit from government subsidies, but these can vary by region and local agricultural policy. They often focus on maximizing efficiency and yields, which can translate into higher production volumes. Conventional milk generally has a larger and more accessible market due to its widespread availability and often at more affordable prices. Conventional systems may have fewer certification and auditing requirements, which could reduce administrative costs.
This aspect should be reviewed, considering what has been described so far, since the choice between organic and conventional systems in milk production implies important considerations both from the perspective of animal welfare, as well as productive and economic aspects, among others. Although organic production may have higher costs, it can also offer market opportunities and higher sales prices. The decision will depend on various factors, such as market demand, local costs, consumer preferences, and the producer's economic goals.
Line 237: … concentrated …
Line 265: … a treatment …
Line 289: … have an elevated …
Line 292: … than on conventional …
Line 230: Review should be Table 3 …
Line 339: ... , immunity prevention ...
Line 341: … on the diet …
Line 350: … pastures …, … the synthesis of …
Line 358: ... differ in the feeding ...
Line 435: ... However, the demonstration ...
Line 449: … and in shelves used for …
Line 452: Table 3: Review the terminology used. … traditional system …, … Traditional Milk …., Intensive Milk … It is not understood that the concept of "Traditional Milk" and "Intensive Milk" could be previously explained, this concept was not found in the scientific literature. Please review.
… Conventional Milk … A definition of this term is missing. Check, thanks
Line 453: .... for the traditional ...
Line 453 -454: Review the terminology used.
Line 464: … in the numbers …, … with respect to, replace by concerning …
Line 464: … had a lower …
Line 511: … yoghurt …
Line 514: … and the risk of prostate …
Line 519 -520: … nutrients in the recommended …
Line 520: … intake, to understand …
Line 539: … of the switch …
Line 541: ... with a lower ...
Line 542: … and a higher …
Line 549: … lowering the chances …
Line 552: … as the risk …
Line 556: … approach to organic …
Line 557: … relies partly on …
Line 578: … for the treatment …
Line 611: … of the climate …, … using pesticides …
Line 630: … intake.
Line 634: … in the composition …
Line 636: … in the abundance …
Comments on the Quality of English Language
Different terms must be reviewed, improved and based on scientific terminology related to production sciences and animal welfare. Furthermore, it is recommended to avoid bias by favouring one system over the others; all systems have advantages and disadvantages; this must be transparent, balanced and comparable.
Reviewer 2 Report
Comments and Suggestions for Authors
See Attach file

Reviewer 3 Report
Comments and Suggestions for Authors
Milk is one of the most valuable products in the food industry, and the organic dairy market is expected to grow due to the advantages of it compared to conventional counterparts persists. This manuscript provides a great overview of the latest research on the production and composition of milk produced using organic agricultural practices.There are several issues in the manuscript that need further improvement.

Round 2
Reviewer 1 Report
Comments and Suggestions for Authors
The requested suggestions, recommendations, and adjustments have been appropriately addressed. The authors have also provided a detailed letter with comments, indicating their acceptance of the proposed improvements to the article. We are grateful to the authors for their cooperation and willingness to consider the suggested changes. The manuscript contains valuable scientific information, has been appropriately revised, and is now ready for publication.
Reviewer 2 Report
Comments and Suggestions for Authors
The revised form of this paper is suitable for publication in Foods
Reviewer 3 Report
Comments and Suggestions for Authors
The author has made enough changes for me to recommend acceptance with minor modifications.
